# Effect of Carbon Nanotubes on the Mechanical, Crystallization, Electrical and Thermal Conductivity Properties of CNT/CCF/PEKK Composites

**DOI:** 10.3390/ma15144950

**Published:** 2022-07-15

**Authors:** Xu Yan, Liang Qiao, Hao Tan, Hongsheng Tan, Changheng Liu, Kaili Zhu, Zhitao Lin, Shanshan Xu

**Affiliations:** 1School of Materials Science and Engineering, Shandong University of Technology, Zibo 255049, China; polaris_xu@foxmail.com (X.Y.); qiaol66@163.com (L.Q.); lin.zhi.tao@163.com (Z.L.); xushanshan@sdut.edu.cn (S.X.); 2Polymeric Nano Materials Laboratory, School of Applied Chemical Engineering, Kyungpook National University, Daegu 41566, Korea; tanhao@naver.com; 3Shandong Qinghe Chemical Technology Co., Ltd., Zibo 255000, China; 15666700706@163.com (C.L.); 18369902273@163.com (K.Z.)

**Keywords:** carbon fiber, carbon nanotube, PEKK, wet power impregnation, laminate

## Abstract

Carbon nanotube/continuous carbon fiber–reinforced poly(etherketoneketone) (CNT/CCF/PEKK) prepreg tapes were prepared by the wet powder impregnation method, and then the prepreg tapes were molded into laminates. The effects of carbon nanotubes on the mechanical properties, conductivity, thermal conductivity and crystallinity of the composites were studied by universal testing machine, thermal conductivity and resistivity tester, dynamic mechanical analyzer (DMA) and differential scanning calorimeter (DSC). The results show that when the content of carbon nanotubes is 0.5 wt% (relative to the mass of PEKK resin, the same below), the flexural strength and interlaminar shear strength of the laminates reach the maximum, which are increased by 15.99% and 18.16%, respectively, compared with the laminates without carbon nanotubes. The results of conductivity and thermal conductivity show that the higher the content of carbon nanotubes, the better the conductivity and thermal conductivity of the material. DSC results show that the addition of CNT promoted the crystallization of PEKK in the material and decreased the cold crystallization of PEKK. DMA results show that the deformation resistance of the material can be improved by adding an appropriate amount of CNT and the bonding between CF and PEKK can be enhanced, while excessive CNT destroys this phenomenon.

## 1. Introduction

Carbon fiber (CF) is one of the most widely used reinforcement materials for high-end applications [1], which has excellent properties such as high temperature resistance, friction resistance, electrical and thermal conductivity and high strength and modulus along the fiber direction. Compared with other fiber reinforcements, carbon fiber–reinforced thermoplastic resin matrix composites (CFRTP) have higher impact resistance, corrosion resistance, high temperature resistance and easy recovery properties, which are unmatched by thermosetting resins [2]. At present, CFRTP has been applied in automotive, aerospace, construction and other fields [3,4], and its application range is gradually expanding.

Poly(etherketoneketone) (PEKK) is a kind of special engineering plastic which has good chemical corrosion resistance, high temperature resistance, radiation resistance, excellent biocompatibility and flame retardance [5,6,7,8]. It can be used as the matrix of carbon fiber (CF)–reinforced resin matrix composites and can be used in aerospace, medical, military and other fields. As a high-performance filler, carbon nanotubes (CNT) can effectively enhance the interfacial adhesion between resin and fiber [9]. Adding a small amount of CNT can significantly improve the properties of the material [10,11,12,13].

Dispersing CNT into composite system mainly includes dispersing CNT into matrix [14] and inserting CNT between layers. To disperse CNTs into the matrix, CNTs are uniformly dispersed into the matrix by means of dry grinding (such as ball milling) or mechanical stirring in suspension and ultrasound, and then the mixture is heated to melt the resin to combine with the fiber. Qiao et al. [15] prepared CNT/CCF/PEKK prepreg tapes by wet powder impregnation. CNT and dispersant polyvinylpyrrolidone (PVP) were first dispersed into absolute ethanol by ultrasound. Peek powder was added and mechanical dispersion continued for 4 h. The prepared suspension was transferred to the dip tank, and CCF was pulled in to impregnate the resin in the dip tank. After melting and plasticizing, CNT/CCF/PEKK prepreg tapes were obtained. The interlaminar insertion method is to insert a CNT film (BP) between the fiber and the substrate and then form it by hot pressing. BP thickness is 50–100 μm. It shows a highly a porous structure formed by the CNT dense network cohesive by van der Waals forces [16]. However, compared with traditional fiber fabrics, BP permeability is reduced by 8–10 times, making it difficult for the resin to penetrate in the processing process. Paula et al. [17] prepared PEI-BP film for CF/PAEK laminates. Compared with BP/CF/PAEK without PEI pad, the load transfer in PEI-BP/CF/PAEK material was better. After PEI-BP was added, the toughness of the material was significantly improved.

CNT is equivalent to a kind of adhesive which can limit the mutual movement between two bonded objects through high interface friction [18]; that is, it improves the interface adhesion between the resin and matrix. Lyu et al. [19] introduced CNT into CF/PEEK composites. The ILSS, flexural strength and modulus of the composites increased by 73.0%, 163.2% and 84.8%, respectively, after CNT was added. The addition of CNT can transform the interface peeling failure into fiber fracture and resin yield, and effectively improve the interface of stress transfer.

In other studies, the common process for introducing CNT into laminates is the film stacking method, which limits the resin immersion in CNT films, resulting in poor wettability. In this work, the CNT/CCF/PEKK prepreg tapes were prepared by the wet powder impregnation method. These prepreg tapes were then prepared as CNT/CCF/PEKK laminates by molding. The flexural properties and viscoelastic behavior of the laminates were analyzed using a universal testing machine and a dynamic mechanical analyzer (DMA), and the effect of CNT on the crystallization of PEKK in the prepreg tape was analyzed using differential scanning calorimetry (DSC).

## 2. Experiment

### 2.1. Materials

PEKK (5 series) resin powder was provided by Shandong Kaisheng New Material Co., Ltd. (Zibo, China). Carbon fibers (T700SC-12K-50C) were produced by Toray Industries, Inc., Tokyo, Japan. Its density is 1.80 g·cm^−3^ and consists of bundles of microfibers (tows) with a single-fiber diameter of 7 μm. CNTs (Array-type) with a diameter of approximately 10–20 nm with a length of 5–50 μm were supplied by Shandong Dazhan Nano Materials Co., Ltd., Zouping, China.

### 2.2. Preparation of Prepreg Tape

In this work, CNT/CCF/PEKK prepreg tapes were prepared by wet powder impregnation. The preparation was divided into three parts: firstly, preparation of impregnation slurry, secondly, preparation of CNT dispersion, and thirdly, prepreg tape preparation.

(1)Preparation of PEKK slurry: Firstly, calculate the required PEKK resin mass and reagent volume according to the requirements of the target product. Prepare PEKK powder and solvent A and solvent B using an analytical balance and measuring cylinder, after which the resin is placed in a container and stirred for at least 1.5 h using a stirrer to ensure that the resin powder is fully infiltrated.(2)Preparation of CNT/PEKK dispersion: Calculate the required CNT mass and PVP mass according to the requirements of the target product. Transfer an appropriate amount of PVP to the beaker, sonicate and disperse for 0.5 h. After that, add CNT and continue sonicating for 1 h to make PVP fully cover CNT. After sonication, mix the CNT dispersion with PEKK slurry and continue stirring for 4 h using a magnetic stirrer.(3)Prepreg tape preparation: Place the fibers in the order of yarn frame, pre-dispersion mold, dipping tank, mold and three-roll calender, turn on the control cabinet to heat the mold and after the set temperature is reached, add slurry to the dipping tank and circulate to ensure that the suspension is not deposited. Turn on the calender, cool the prepared prepreg tape and roll it. Figure 1 shows the flow chart of CNT/CCF/PEKK prepreg tape preparation.

### 2.3. Laminate Preparation

Compared with the prepreg tape, the distribution of resin and fiber in the laminate is more uniform and the porosity is lower. Experimentally, CNT/CCF/PEKK laminates were prepared by molding the prepared prepreg tape in 0°, 0/90°, and woven manner. Firstly, the prepreg tapes were cut into laminates suitable for the mold size and laid in the mold. After that, the mold was placed in the center of the press plate of the flat vulcanizer, and when it was heated up to above the glass transition temperature of PEKK, the hydraulic pump of the flat vulcanizer was activated and pressurized to 2 MPa and then exhausted 7 times when the temperature reached 340 °C. After that, the pressure was kept for 15 min according to the experimental requirements, and the mold was cooled to room temperature and demolded to obtain CNT/CCF/PEKK laminate. The preparation process of CNT/CCF/PEKK laminate is shown in Figure 2.

### 2.4. Measurements

#### 2.4.1. Flexural Strength Test

The prepared laminates were cut to the standard required size by using a universal sample-making machine according to ISO 178-2010 standard. Then, the flexural strength of the prepared laminate samples was tested using a universal testing machine, type 5969, manufactured by Instron, Norwood, MA, USA. Before the test, the universal prototype was used to cut the laminates. The sample size is shown in Figure 3, where h is the thickness of the samples, l is the length of the samples (not less than 20 times the thickness), and b is the width of the samples. During the test, the span thickness ratio was set as (16 ± 1): 1, the test speed as 10 mm/min, and 5 effective data for each group were taken.

#### 2.4.2. In-Surface Thermal Conductivity

According to the ISO 22007-2-2008 standard, the thermal conductivity (λ) of the laminate was tested using a thermal conductivity tester. Two small samples of 2 × 5 cm^2^ were made from the laminates before the test. The samples were wiped clean with anhydrous ethanol, dried in a vacuum drying oven and placed in a standard environment for more than 24 h. Before starting the test, the thermal conductivity meter should be opened and preheated for more than 15 min, and the stability of the thermal conductivity meter should be checked by using a standard test block. Two small samples were clamped on the upper and lower sides of the heat transfer probe. The test temperature was (25 ± 2) °C, the test power is 0.25 W, and the sampling time was 160 s. Each group of samples was repeated 5 times, and the average value was taken.

#### 2.4.3. Electrical Conductivity

According to T/CSTM 00252-2020 standard issued by Zhongguancun material testing technology alliance, the electrical conductivity (*σ*) of the laminate was tested using a one-type four-probe resistivity tester. The sample thickness was input into the system, and the sample conductivity was recorded by pressing the probe on the sample surface with the knob, and each group of samples was tested 5 times and the average value was taken.

#### 2.4.4. Interlaminar Shear Strength

The plates were tested for interlaminar shear strength by the short-beam method using a universal material testing machine according to the standard ISO 14130-1997. First, the plate was cut into specimens of width 2–3 h and length 5 h + 10 mm using a universal sample-making machine, where h is the thickness. Short-beam method interlaminar shear strength calculation formula is as in formula (1):(1)$$\tau_{m}=\frac{3\;\times \;P}{4\;\times \;bh}$$
where *τ_m_* is the interlaminar shear strength (MPa); *P* is the breaking load (N); *b* is the specimen width (mm); *h* is the specimen thickness (mm).

#### 2.4.5. DSC

The crystalline melting behaviors of CNT/CCF/PEKK, CCF/PEKK, CNT/PEKK plates and PEKK were analyzed and tested using a differential scanning calorimeter with a sample mass of 6–7 mg. Firstly, the temperature was ramped up to 360 °C at 10 °C/min and thermostated for 2 min and then at 10 °C/min cooled to 80 °C, held at constant temperature for 2 min, and then at 10 °C/min heated to 360 °C. The crystallinity of the material was calculated from the enthalpy change recorded by the instrumental test, where the crystallinity of the composite material in the presence of cold crystallization was calculated by the formula (2) as follows [20]:(2)$${Χ}_{c}=\frac{\left|\Delta H_{m}-\Delta H_{cc}\right|}{\Phi_{PEKK}\Delta H_{m}^{0}}$$
where *Χ_c_* is the crystallinity of the composite, Δ*H_m_* is the melting enthalpy of the composite, Δ*H_cc_* is the cold crystallization enthalpy of the composite, *Φ_PEKK_* is the content of *PEKK* resin in the composite, and $\Delta H_{m}^{0}$ is the melting enthalpy of PEKK when it is fully crystallized, with a value of 130 J/g.

#### 2.4.6. DMA

The effect of CNTs on the dynamic mechanical behavior of CNT/CCF/PEKK laminates was tested and characterized according to ISO/6721-1-2011 using the single-cantilever deformation mode of a dynamic mechanical analyzer. The laminates were cut into sample strips of 6–10 mm in width and 35 mm in length before the test. For the test, the heating rate was 5 °C/min, the temperature range was 80–250 °C, and the frequency was set to 1 HZ.

## 3. Results and Discussion

### 3.1. Flexural Strength

The flexural stress–strain curves of laminates with different CNT contents at 65 wt% CF content are shown in Figure 4. As shown in this figure, the flexural strength and flexural modulus of the laminates increase and then decrease with the increase in CNT content. The flexural strength reaches a maximum of 1494 MPa when the CNT content is 0.5 wt%. Compared with the laminate without CNT addition, the flexural strength increased by 15.99%. This indicates that the addition of CNT improves the interface between the laminate layers and increases the bond strength between the layers. When the CNT content reached 1.0 wt%, the flexural strength and modulus of the laminate show a decreasing trend. This is because the excess CNT agglomerates inside the material and distributes unevenly inside the laminate, and the layers are more likely to debond and fail at the CNT agglomeration.

### 3.2. In-Surface Thermal Conductivity and Electrical Conductivity

Figure 5 illustrates the λ for laminates with different CNT contents at 65 wt% CF. In the figure, the λ increases by 31.47% and 33.64% for laminates with CNT content of 0.5 wt% and 1.0 wt%, respectively, compared to laminates without CNT addition. The heat in the materials relies on phonon transport [21], and the varying lengths of molecular chains in PEKK resins, the presence of a large number of amorphous regions, and the irregularly connected entanglement of PEKK chain segments limit the phonon transfer, in contrast to the excellent λ and ballistic transport properties that CNT has. The CNT dispersed in the laminate refines the phonon transport network, which in turn increases the λ of the material [22].

In the fields of aerospace and electronic communication, composite materials often need to have some electrical conductivity in order to be used as lightning strike protection materials and shielding materials. Therefore, it is meaningful to investigate the σ of the CNT/CCF/PEKK composites. Figure 6 shows the effect of different CNT contents on the σ of the material at a CF content of 65 wt%. In this figure, the *σ*_0_ and *σ*_90_ of the material increase with the increase in CNT, and the *σ*_0_ and *σ*_90_ of the material with CNT content of 1.0 wt% increased by 213% and 148%, respectively, compared to the material without CNT addition. This is because CNT has good electrical conductivity, and after adding CNT, a perfect electron transfer network is formed in the CNT/CCF/PEKK materials. The higher the CNT content, the better the electron transfer network.

### 3.3. Interlaminar Shear Strength

Figure 7 shows the relationship between different CNT contents and ILSS of the CNT/CCF/PEKK laminates. In the figure, the ILSS of the laminates with 0.5 wt% and 1.0 wt% CNT increased by 18.16% and 8.29%, respectively, compared to the laminates without CNT addition, which is due to the penetration of CNT into the internal layers of the laminates, resulting in an increase in the degree of interlayer compactness and an increase in the resistance to interlayer movement. In addition, the ILSS of the 1.0 wt% CNT-content laminate was 8.34% lower than that of the 0.5 wt% CNT laminate. This may be due to the fact that a large amount of CNT agglomerates in the laminate and breaks the degree of interlaminar bonding at the agglomerates, causing the laminate to peel off the interlaminar by stress concentration when subjected to shear force. In the study of Yousefi et al. [23], it was found that excessive CNT also led to the decrease in ILSS. They attributed this phenomenon to the high voids in the composites.

### 3.4. DSC Analysis

Figure 8 shows the first ramp-up curves of DSC for the prepreg tapes with different CNT contents (relative to PEKK resin, below) at 65 wt% CF content. As shown in this figure, the addition of CNT has almost no effect on the *T_g_* of CNT/CCF/PEKK. This means that there is almost no chemical bonding between CNT and PEKK substrates [24]. Compared with the CNT/PEKK composite without the addition of CF, a crystallization peak, i.e., a cold crystallization peak, appeared during the warming process of CNT/CCF/PEKK. This is due to the large cooling rate during the preparation of the composites, which inhibits the chain reorganization that occurs in the melt during cooling and hinders the formation of the crystalline structure with the presence of a large amount of the amorphous phase [25]. When the material is heated again, the amorphous phase gains energy to rearrange, causing cold crystallization to occur when the material is heated. The Δ*H*_cc_ of the material decreases and the cold crystallization peak temperature (*T*_cc_) shifts toward the lower temperature after the CNT addition. This indicates that the addition of CNT promotes the crystallization of PEKK at a high cooling rate.

Figure 9 shows the cooling curves of the CNT/CCF/PEKK composites with different CNT contents, and in this figure, no obvious crystallization peaks appear in the cooling curves of the composite without the addition of CNT, while the crystallization peaks of the composites are obvious after the addition of CNT, and the temperature of the crystallization peaks moves toward the high temperatures with the increase of the CNT content. This may be due to the fact that the CF surface used in this paper has a high surface energy, which limits the growth of PEKK transverse crystals [26] so that PEKK crystals can only grow along the fiber direction, and at high fiber content, the spacing between CFs is small, which limits the growth of the PEKK spherical crystals. After adding CNT, CNT forms a network inside the material, and PEKK can crystallize along the CNT network, making the material easier to crystallize, with obvious crystallization peaks and moving toward higher temperatures with increasing CNT content.

Figure 10 shows the secondary temperature rise curves of the CNT/CCF/PEKK composites after eliminating the thermal history. In this figure, the cold crystallization peak of the added CNT material disappears, while the cold crystallization peak of the unadded CNT material is still clearly present, and the crystallinity of the material after adding CNT is significantly higher than that of the unadded CNT composite. This is because the high CF content restricts the movement of PEKK chain segments and leads to the imperfect crystallization of PEKK. After adding CNT, PEKK crystallizes more perfectly and fully, the amount of amorphous PEKK decreases, and the cold crystallization disappears after adding CNT and the *T_m_* of the material moves toward the high temperatures. This indicates that although CNT promotes the crystallization of PEKK in the composite, it also restricts the movement of PEKK chain segments. In addition, as shown in Table 1, the higher the CNT content, the faster the crystallization rate of PEKK, and when the CNT content is 0.5 wt%, the crystallinity of PEKK is higher. This is probably because the higher the CNT content, the higher the crystalline nucleus, but this causes the PEKK spherical crystal size to decrease, resulting in a higher PEKK crystallinity [27]. In addition, too much CNT restricts the growth space of the PEKK crystal and hinders the growth of PEKK crystal.

### 3.5. DMA

The storage modulus (*E*′*)* reflects the ability of a material to resist deformation. Figure 11 shows the *E*′-temperature curves of laminates with different CNT contents at 65 wt% of CF. In this figure, the initial *E*′ of the laminate is highest when 0.5 wt% (relative to the mass of PEKK, same below) CNT is added to the laminate. This is because the addition of a moderate amount of CNT can improve the deformation resistance of the material and increase the bonding force between CF and PEKK. The initial *E*′ of the laminate is lower when 1.0 wt% CNT is added, which is because the excessive amount of CNT leads to an obvious agglomeration phenomenon, and the agglomeration breaks the interface between CF and PEKK. In addition, the *E*′ drop is greater for the laminate without CNT addition, suggesting that the addition of CNT can increase the potential of the material for long-term use.

The loss factor (tan*δ*) is the ratio of loss modulus (*E*″) to *E*′. Figure 12 shows the tan*δ*-temperature curves of the three laminates. In this figure, the tan*δ* peak temperature is lower for the laminate without CNT compared to the laminates with CNT, which indicates that the addition of CNT increases the resistance of the laminate to high temperature deformation. The tan*δ* peak can reflect the degree of bonding between CF and PEKK, and the lower the tan peak, the better the bonding between the resin and CF. Compared with the laminate without CNT and the laminate with 1.0 wt% CNT, the tan*δ* peak is lower for the laminate with 0.5 wt% CNT content, which indicates that the bonding between CF and PEKK is better in the laminate with 0.5 wt% CNT content. In addition, the peak temperature of tan*δ* is generally considered to be the *T_g_* of the material. In the figure, after adding CNT, the *T_g_* of the laminate moves to the high temperatures, which indicates that CNT can improve the service temperature limit of the laminate.

## 4. Conclusions

In this work, CNT/CCF/PEKK prepreg tapes were prepared by wet powder impregnation, and the prepreg tapes were formed into laminates by molding. The effects of CNT on the mechanical, electrical and thermal conductivity and the melting crystallization behavior of the composites were studied. The main conclusions are as follows:

(1)An appropriate amount of CNT can improve the adhesion between CF and PEKK and increase the flexural strength and ILSS of CNT/CCF/PEKK laminates. In this study, the 0.5 wt% CNT-content laminate has higher flexural strength and ILSS. When the CNT content increases to 1.0 wt%, the flexural strength and ILSS of the laminate decrease.(2)CNT can form a perfect transmission network of phonons and electrons inside the material. In the range of 0–1.0 wt%, the higher the CNT content, the better the conductivity and thermal conductivity of the material.(3)CNT can form a perfect crystallization network in the material, and PEKK can crystallize at a higher fiber content. In addition, CNT plays the role of heterogeneous nucleation in the material. In the range of 0–1.0 wt%, the higher the CNT content, the more crystalline nuclei, and the faster the PEKK crystallization rate. When CNT content is 0.5 wt%, PEKK crystallinity is higher, and when CNT content increases to 1.0 wt%, PEKK crystallinity decreases.(4)An appropriate amount of CNT can improve the storage modulus of the laminates, the adhesion between CF and PEKK, and *T_g_*. In this work, 0.5 wt% CNT-content laminates have higher storage modulus, lower tan*δ* peaks and higher *T_g_*.

In a word, the 0.5 wt% CNT-content laminate has more application potential.

## Figures and Tables

**Figure 1 materials-15-04950-f001:**
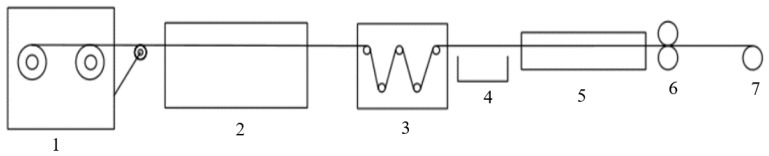
Production and preparation flow chart of CNT/CCF/PEKK prepreg tapes. 1: yarn frame; 2: yarn-spreading device; 3: impregnation device; 4: recycling tank; 5: melt-plasticizing molds; 6: calendering equipment; 7: winding device.

**Figure 2 materials-15-04950-f002:**
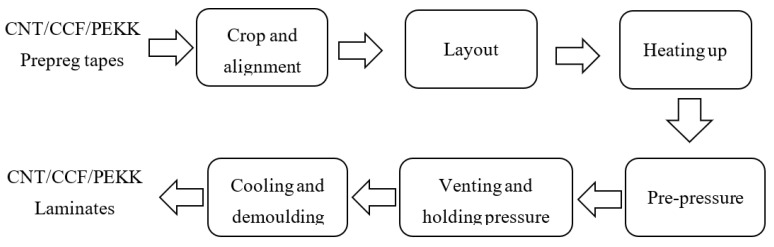
Preparation flowchart of CNT/CCF/PEKK laminate.

**Figure 3 materials-15-04950-f003:**
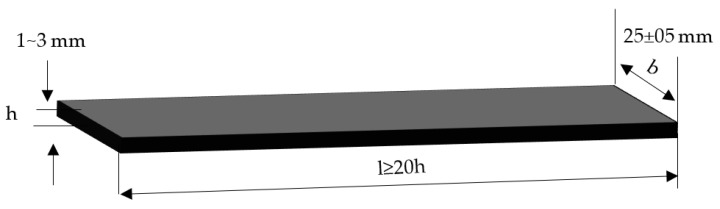
Dimension diagram of flexural specimen.

**Figure 4 materials-15-04950-f004:**
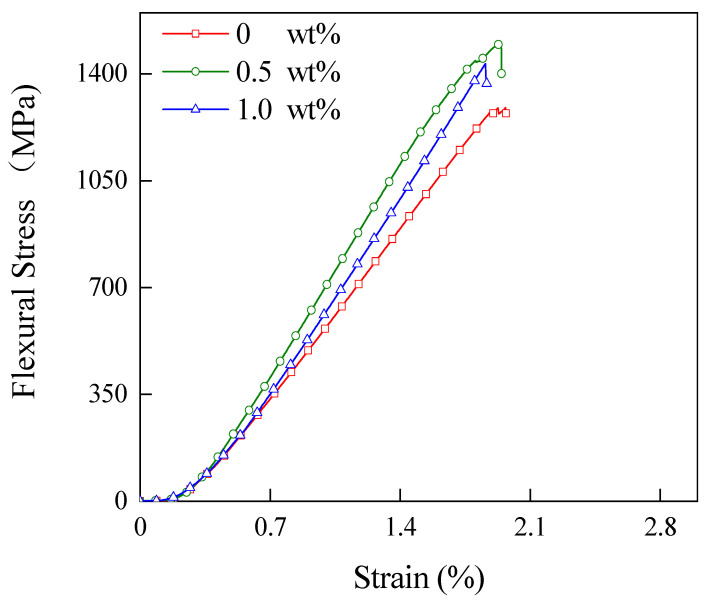
The flexural stress–strain curves of the composites with different CNT contents.

**Figure 5 materials-15-04950-f005:**
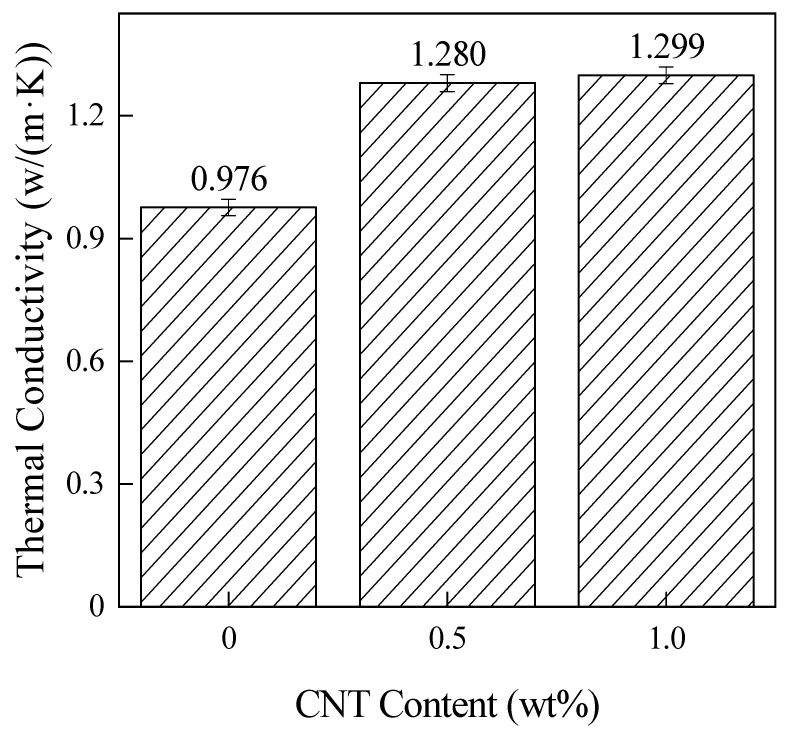
In-surface thermal conductivity of the composites with different CNT contents.

**Figure 6 materials-15-04950-f006:**
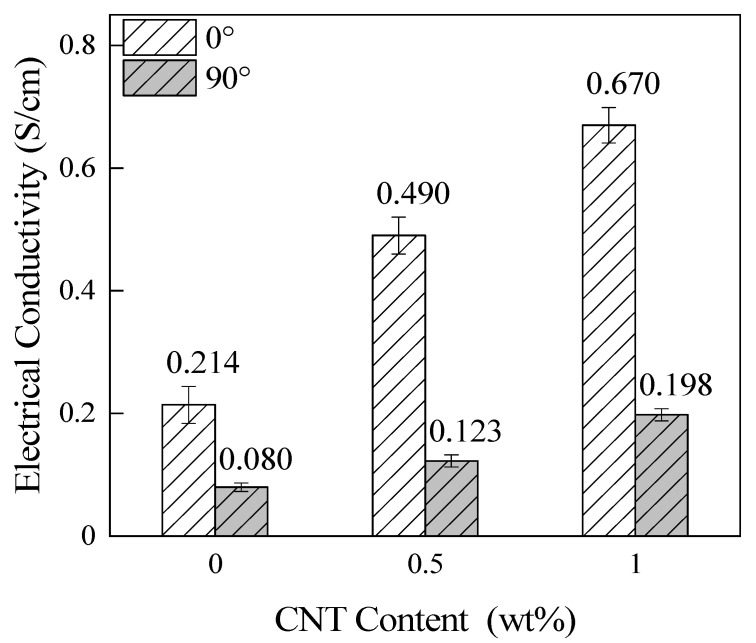
Electrical conductivity of the composites with different CNT contents in different CF directions.

**Figure 7 materials-15-04950-f007:**
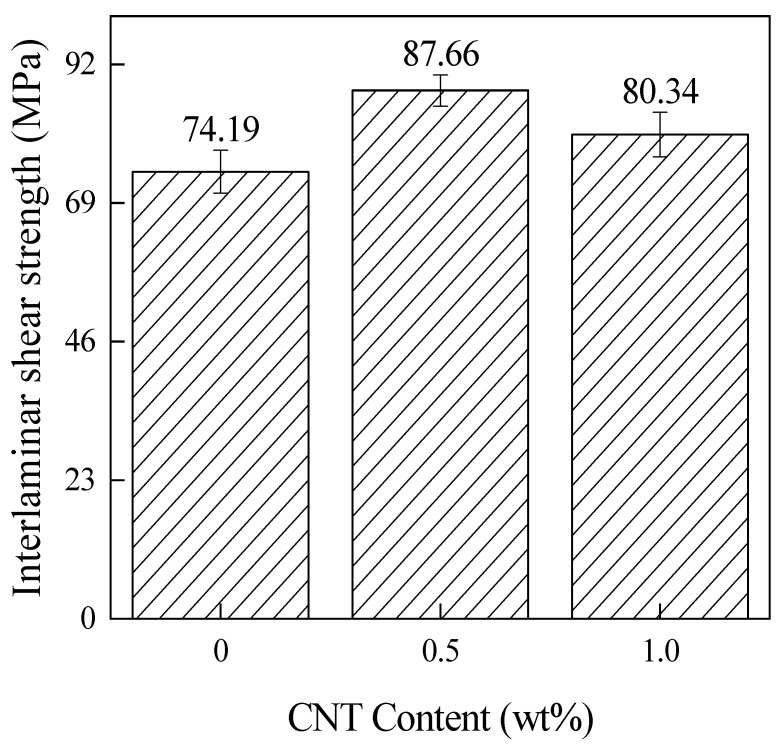
Interlaminar shear strength of composites with different CNT contents.

**Figure 8 materials-15-04950-f008:**
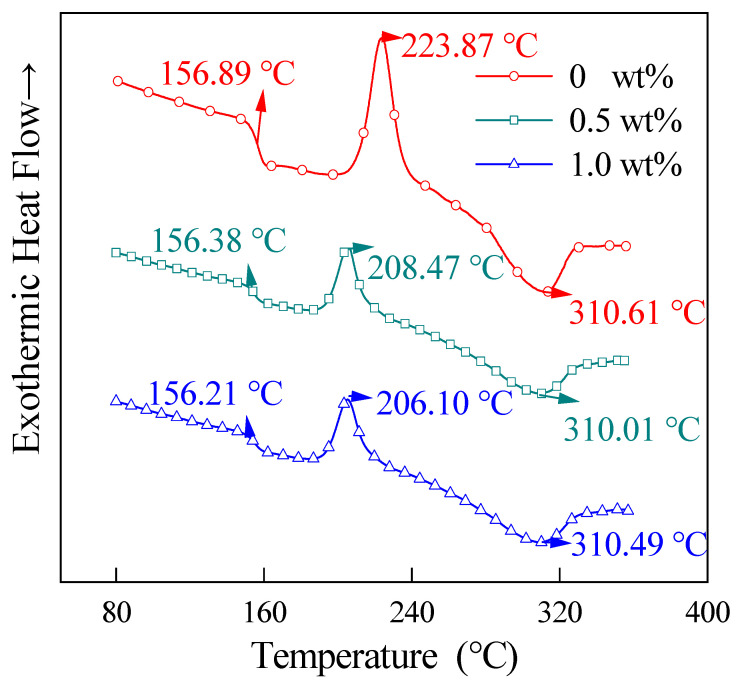
The DSC first heating curves of composites with different CNT contents.

**Figure 9 materials-15-04950-f009:**
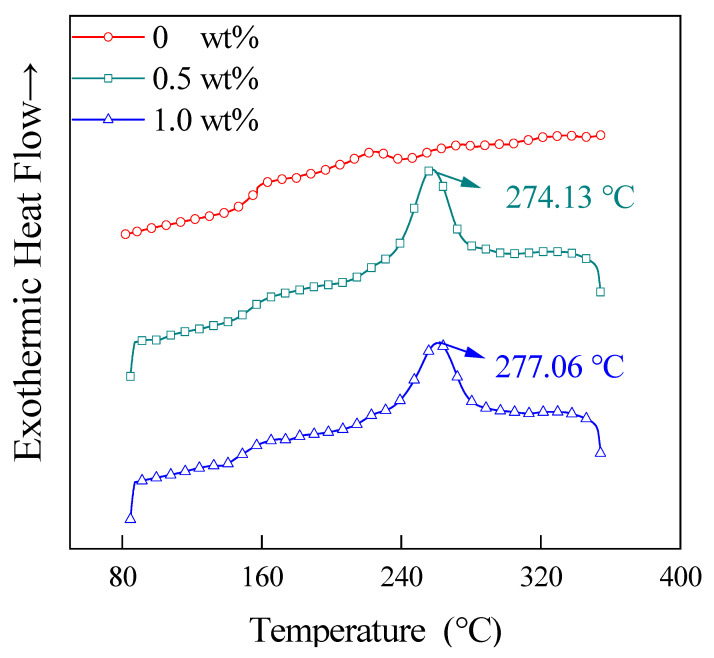
The DSC cooling curves of composites with different CNT contents.

**Figure 10 materials-15-04950-f010:**
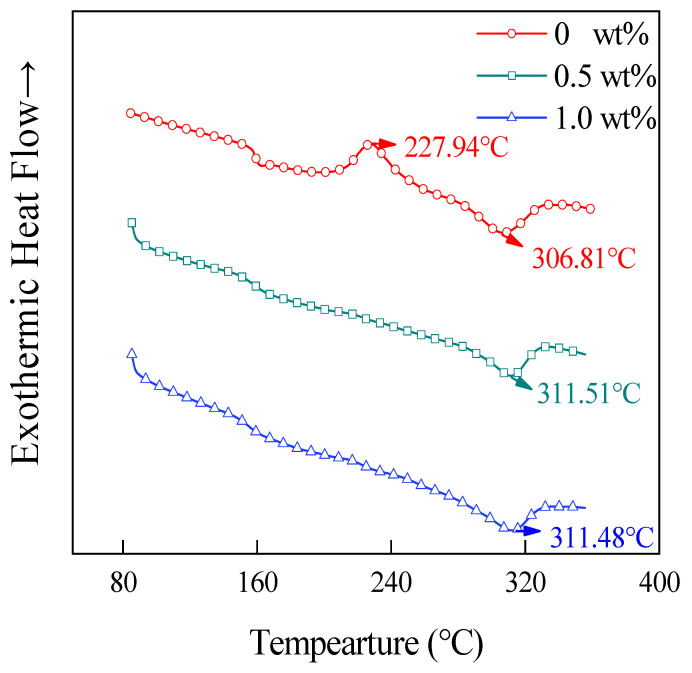
The DSC second heating curves of composites with different CNT contents.

**Figure 11 materials-15-04950-f011:**
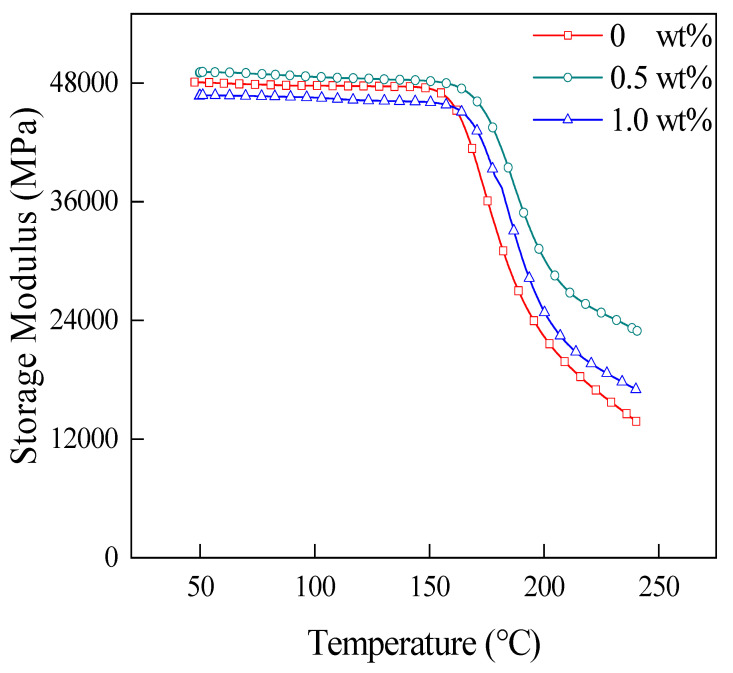
The *E*′-temperature curves of laminates with different CNT contents.

**Figure 12 materials-15-04950-f012:**
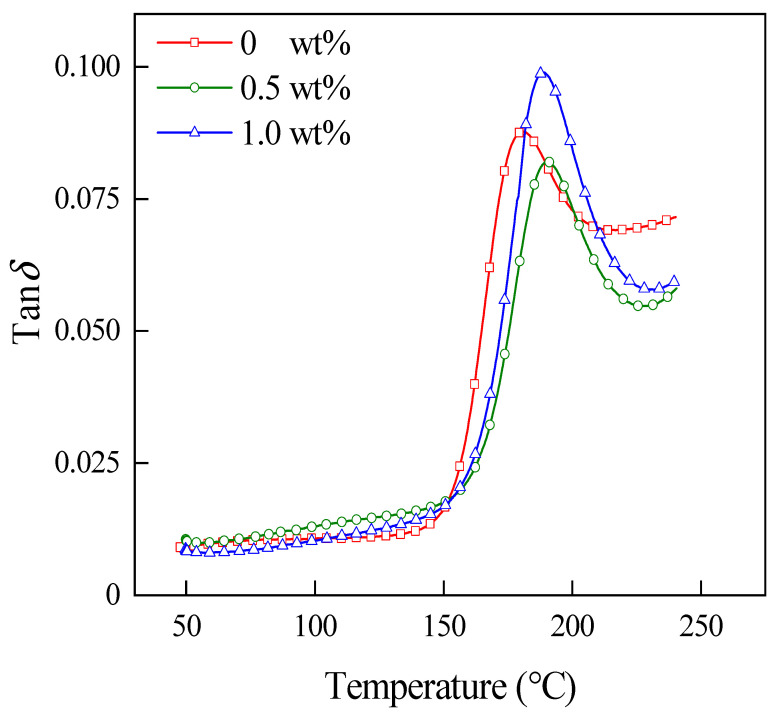
The tan*δ*-temperature curves of laminates with different CNT contents.

**Table 1 materials-15-04950-t001:** *T_cc_*, *T_m_*, *T_c_*, Δ*H_m_*, Δ*H_cc_*, *Χ_c_* and *t*_1/2_ of prepregs with different CNT contents.

CNT Contents (wt%)	*T_cc_* (°C)	*T_m_* (°C)	*T_c_* (°C)	Δ*H_m_* (J/g)	Δ*H_cc_* (J/g)	*Χ_c_* (%)	*t*_1/2_ (min)
0	227.94	306.81	-	7.45	4.25	2.46	-
0.5%	-	311.51	256.55	6.09	-	4.68	3.26
1.0	-	311.48	260.67	6.04	-	4.64	3.00

## Data Availability

The data used in this work has been properly cited and reported in the main text.

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
