# Peer review of "Effect of Carbon Nanotubes on the Mechanical, Crystallization, Electrical and Thermal Conductivity Properties of CNT/CCF/PEKK Composites"

_materials, 2022, doi:10.3390/ma15144950_

Round 1
Reviewer 1 Report
Reviewers' comments:
In the manuscript, the authors reported the preparation of carbon nanotube/continuous carbon fiber reinforced Polyether-ketone-ketone (CNT/CCF/PEKK) prepreg tapes through the wet powder impregnation method, and then the preparation of prepreg tapes into laminates by molding. Generally, current work is well carried out. However, the authors should try to emphasize better the importance of current manuscript in order to attract the readership of this journal. Besides, the publication of the work in this reputable journal can be justified although the novelty is not really emphasized. Besides, the research work in this manuscript can be accepted and published in this high reputed journal after the authors consider the following major points.
TECHNICAL PART
1. The English language needs significant attention and substantial improvement as several sentences are not quite clear.
ABSTRACT PART
1. The authors should highlight Hypothesis, Experiments and Findings. The preferred format for the Abstract should be used in order to attract the readership of this journal.
2. The Abstract also can be accompanied by numerical data/quantitative information to support the statements.
INTRODUCTION PART
1. The introduction part is too casual, not well written and don’t have the flow of explaination. Please revise the whole introduction part.
2. How this research work is different from other published work?
3. Please highlight the novelty of this study properly.
RESULTS AND DISCUSSION PART
1. The contribution with respect to other similar works appeared in literature (compare results in a table) and the possible future commercial application of the carbon nanotube/continuous carbon fiber reinforced Polyether-ketone-ketone (CNT/CCF/PEKK) prepreg tapes should be point out.
CONCLUSION PART
1. Manuscripts published in this journal must explain the significant advances provided in approaches and understanding compared to previous literature, and/or demonstrate convincingly potential in new applications. The conclusions of your paper are especially important for this. Therefore, please try to sharpen this further. The optimal conclusion should include: A summary of your findings, A synopsis of your new concepts and innovations, A brief restatement of your hypotheses, Your vision for future work.
2. There are many repeated sentences and not proper words. Please revise.
Author Response
Dear reviewer:
I am very grateful to your comments for the manuscript. According with your advice, we amended the relevant part in manuscript. Some of your questions were answered below.
TECHNICAL PART
- The English language needs significant attention and substantial improvement as several sentences are not quite clear.
R: We have carefully considered the comments of the reviewers, checked and revised the grammar and technical words of the full text
ABSTRACT PART
- The authors should highlight Hypothesis, Experiments and Findings. The preferred format for the Abstract should be used in order to attract the readership of this journal.
2.The Abstract also can be accompanied by numerical data/quantitative information to support the statements.
R: We have carefully considered the comments of reviewers and revised this part
INTRODUCTION PART
- The introduction part is too casual, not well written and don’t have the flow of explaination. Please revise the whole introduction part.
- How this research work is different from other published work?
- Please highlight the novelty of this study properly.
R: We have carefully considered the reviewers' suggestions, and revised and added content to this part.
RESULTS AND DISCUSSION PART
- The contribution with respect to other similar works appeared in literature (compare results in a table) and the possible future commercial application of the carbon nanotube/continuous carbon fiber reinforced Polyether-ketone-ketone (CNT/CCF/PEKK) prepreg tapes should be point out.
R: We have carefully considered the reviewers' suggestions, and revised and added content to this part.
CONCLUSION PART
- Manuscripts published in this journal must explain the significant advances provided in approaches and understanding compared to previous literature, and/or demonstrate convincingly potential in new applications. The conclusions of your paper are especially important for this. Therefore, please try to sharpen this further. The optimal conclusion should include: A summary of your findings, A synopsis of your new concepts and innovations, A brief restatement of your hypotheses, Your vision for future work.
R: We have carefully considered the comments of the reviewers and revised the conclusion.
- There are many repeated sentences and not proper words. Please revise.
R: We have carefully considered the comments of the reviewers, checked and revised the grammar and technical words of the full text.
Thank you again for your professional and meticulous advice.
Best wishes,
Xu Yan
Reviewer 2 Report
This paper concludes that the laminate 0.5 wt% CNT(relative to the mass of PEKK) has higher flexural strength than the laminate without added CNT and the laminate with 1.0 wt% CNT. Also, the higher the CNT content, the faster the crystallization rate of composite PEKK, with 0.5 wt% CNT content composites having higher crystallinity. The results showed that CNT can effectively improve the electrical and thermal conductivity of the composites. The higher the CNT content, the better the electrical and thermal conductivity of the composites.
Overall this work can be considered for publication. However, to support the conclusion, the following issues should be addressed.
1. The conclusion of highest flexural strength achieved with 0.5 wt% must be strengthened by supporting data. More samples should be tested: at least three samples/testes for each composite. For example, the current data shown in Figure 3 would not be reliable to draw the conclusion because the difference in flexural stress/strength between composites is small.
2. I have to suggest to show SEM or TEM images of the composites to show the micro/nano structures as well as support for the conclusion of the mechanisms on how CNT affect the properties of the composites
3. I suggest to proofread the manuscript carefully as important information is misleading, e.g. Figure 10 shows “Strongth Modulus”, is that correct, or it should be storage modulus?
Author Response
Dear reviewer:
I am very grateful to your comments for the manuscript. According with your advice, we amended the relevant part in manuscript. Some of your questions were answered below.
1.The conclusion of highest flexural strength achieved with 0.5 wt% must be strengthened by supporting data. More samples should be tested: at least three samples/testes for each composite. For example, the current data shown in Figure 3 would not be reliable to draw the conclusion because the difference in flexural stress/strength between composites is small.
R: We checked the relevant data. During the test, we take 5 valid data. Because CCF is the main stress bearer in CNT/CCF/PEKK, the mechanical properties of laminates do not change greatly after CNT is added.
2.I have to suggest to show SEM or TEM images of the composites to show the micro/nano structures as well as support for the conclusion of the mechanisms on how CNT affect the properties of the composites
R: We considered the reviewer's suggestion and found it very pertinent. Since CNT/CCF/PEKK laminates are difficult to pull breaking and press breaking under the action of universal testing machine, we only analyze the submicroscopic morphology of the tensile section of cnt/ccf/pekk prepreg, and will reflect in detail the mechanism of CNT's influence on the composite interface in the next work.
3.I suggest to proofread the manuscript carefully as important information is misleading, e.g. Figure 10 shows “Strongth Modulus”, is that correct, or it should be storage modulus?
R:After checking, it is indeed our negligence that led to the wrong use of technical words.
Thank you again for your professional and meticulous advice.
Best wishes,
Xu Yan
Reviewer 3 Report
To improve the quality of the manuscript following changes need to incorporate in the manuscript.
1. Introduction part of the manuscript need to contain the importance of carbon nanotubes and continuous carbon fibers reinforced composites for industrial applications.
2. Detailed composites preparation with real time images need to add in the laminate preparation section of the manuscript.
3. For in surface thermal conductivity, electrical conductivity, and remaining other tests required to add ASTM standards.
4. Schematic a or specimen image with dimensions for flexural test is required.
5. Inter-laminar shear strength has been decreased for 1 wt.% of CNT reinforced composites, justify technically with proper references.
6. Justify the increase in thermal conductivity with increased content of CNT with suitable references in the manuscript.
Author Response
Dear reviewer:
I am very grateful to your comments for the manuscript. According with your advice, we amended the relevant part in manuscript. Some of your questions were answered below.
- Introduction part of the manuscript need to contain the importance of carbon nanotubes and continuous carbon fibers reinforced composites for industrial applications.
R:We have carefully considered the reviewers' suggestions, and revised and added content to this part.
- Detailed composites preparation with real time images need to add in the laminate preparation section of the manuscript.
R:We have considered the reviewer's suggestions, but since the preparation process involves other people's patents, we cannot take photos during the preparation process.
- For in surface thermal conductivity, electrical conductivity, and remaining other tests required to add ASTM standards.
R:According to the reviewers' suggestions, we have modified this part and added relevant standards.
- Schematic a or specimen image with dimensions for flexural test is required.
R:According to the reviewers' suggestions, we have modified this part.
- Inter-laminar shear strength has been decreased for 1 wt.% of CNT reinforced composites, justify technically with proper references.
R:According to the reviewers' suggestions, we have modified this part and added relevant references.
- Justify the increase in thermal conductivity with increased content of CNT with suitable references in the manuscript.
R:According to the reviewers' suggestions, we have modified this part and added relevant references.
Thank you again for your professional and meticulous advice.
Best wishes,
Xu Yan
Round 2
Reviewer 2 Report
The response is satisfactory
Reviewer 3 Report
Authors have modified the manuscript as per reviewers comments.
Manuscript can be accepted for publication in its present form.